# The Impact of Smoking on Clinical Results Following the Rotator Cuff and Biceps Tendon Complex Arthroscopic Surgery

**DOI:** 10.3390/jcm10040599

**Published:** 2021-02-05

**Authors:** Jan Zabrzyński, Gazi Huri, Maciej Gagat, Łukasz Łapaj, Alper Yataganbaba, Dawid Szwedowski, Mehmet Askin, Łukasz Paczesny

**Affiliations:** 1Department of General Orthopaedics, Musculoskeletal Oncology and Trauma Surgery, University of Medical Sciences, 61-701 Poznan, Poland; esperal@o2.pl; 2Department of Orthopaedics, Orvit Clinic, Citomed Healthcare Center, 87-100 Torun, Poland; drpaczesny@orvit.pl; 3Department of Pathology, Faculty of Medicine, Nicolaus Copernicus University in Torun, 87-100 Torun, Poland; 4Orthopaedics and Traumatology Department, Hacettepe Universitesi, Ankara 06-532, Turkey; gazihuri@yahoo.com (G.H.); alperyataganbaba@gmail.com (A.Y.); mehmetaskin7@gmail.com (M.A.); 5Department of Histology and Embryology, Faculty of Medicine, Nicolaus Copernicus University in Torun, 87-100 Torun, Poland; mgagat@cm.umk.pl; 6Orthopaedic Arthroscopic Surgery International (OASI) Bioresearch Foundation, 20-133 Milan, Italy; dszwedow@yahoo.com

**Keywords:** massive rotator cuff tears, LHBT, biceps, rotator cuff, tenodesis, tenotomy, tendinopathy, arthroscopy, shoulder, smoking

## Abstract

The purpose of this study was to investigate the association of smoking and functional outcomes after arthroscopic treatment of complex shoulder injuries: rotator cuff tears (RCTs) with biceps tendon (LHBT) tears. This retrospective case-control study has been conducted on a cohort of patients who underwent shoulder arthroscopy between 2015 and 2017 due to complex injury treatment. The outcomes were assessed using the American Shoulder and Elbow Surgeons Score (ASES), the University of California at Los Angeles (UCLA) Shoulder Score, need for non-steroid anti-inflammatory drugs (NSAIDs) consumption and the visual analog scale (VAS). Complications and changes in smoking status were also noted. A cohort of 59 patients underwent shoulder arthroscopy, due to complex LHBT pathology and RCTs, and were enrolled in the final follow-up examination; with mean duration of 26.03 months. According to smoking status, 27 of patients were classified as smokers, and the remaining 32 were non-smokers. In the examined cohort, 36 patients underwent the LHBT tenotomy and 23 tenodesis. We observed a relationship between smoking status and distribution of various RCTs (*p* < 0.0001). The mean postoperative ASES and UCLA scores were 80.81 and 30.18 in the smoker’s group and 84.06 and 30.93 in the non-smoker’s group, respectively. There were no statistically significant differences in pre/postoperative ASES and postoperative UCLA scores between smokers and non-smokers (*p* > 0.05). The VAS was significantly lower in the non-smokers’ group (*p* = 0.0021). Multi-tendon injuries of the shoulder are a serious challenge for surgeons, and to obtain an excellent functional outcome, we need to limit the negative risk factors, including smoking. Furthermore, there is a significant association between smoking and the occurrence of massive rotator cuff tears, and the pain level measured by the VAS. Simultaneous surgical treatment of RC and LHBT lesions in the smoker population allowed us to obtain the functional outcomes approximated to non-smokers in the long-term follow-up. Of course, we cannot assert that smoking is the real cause of all complications, however, we may assume that this is a very important, negative factor in shoulder arthroscopy.

## 1. Introduction

Tobacco smoking is a common, crucial negative factor in oncological, pulmonary and cardiovascular diseases. Furthermore, tobacco smoking has an undeniable impact on the musculoskeletal system and is a detrimental factor for poor postoperative outcomes after various surgical procedures [1,2,3,4,5]. However, only a few studies analyzed the link between shoulder surgery and smoking, though there is still a lack of adequate data in this area [6,7,8,9,10,11]. Smoking has a negative influence on cells, tissues, vascularity, and metabolism. Its multidimensional effect in orthopedic surgery results in a delayed union, or non-union of fractures, increased risk of infection, impaired wound healing and increased rate of additional complications [12,13].

A significant group of the population suffers from shoulder pain due to acute or chronic tendon injuries, which are becoming a considerable cause of work disability. Various tendon disorders may appear simultaneously in different localizations of the shoulder [14]. Rotator cuff tendinopathy and tears (RCTs) are the most common ones among them. They are usually associated with the long head of the biceps tendon (LHBT) pathology, superior labrum anterior to posterior (SLAP) injuries, subacromial impingement syndrome and acromioclavicular joint (ACJ) disorders [15,16,17,18]. After the supraspinatus tendon, the most common injured structure of the rotator cuff (RC) complex, biceps tendon is an element of compensation of the abnormal forces and tears of which predispose to subsequent instability, and further subscapularis tendon tears. Kelly et al. revealed LHBT disorders with various co-existing shoulder pathologies, such as RCTs, resulting in instability of the shoulder and subacromial impingement [19]. Furthermore, the massive RCTs can lead to accelerated omarthrosis and shoulder dysfunction. Complex and multi-tendon shoulder injuries significantly complicate the process of diagnosis, treatment, and rehabilitation.

A few studies deal with the influence of smoking on isolated RC or labrum surgery, however none of them has investigated complex, heterogenous injuries of RC and LHBT, which are actually more commonly encountered during arthroscopic operations than previously thought [7,10,20,21,22]. Moreover, the results of these studies are inconsistent. Mallon et al. and Balyk et al. found a statistically significant negative association between smoking and clinical outcomes after RC surgery [20,23]. On the contrary, Prasad et al. were unable to prove such a relationship [24]. A complex injury of the shoulder is difficult to treat, due to the need of more extensive surgery. Furthermore, tendinous tissue exposed to tobacco smoke may have an impaired healing biology, resulting in poor clinical outcomes [21]. Rotator cuff and biceps tendons have areas of limited blood supply known as “watershed areas” [25]. These hypovascular areas are more sensitive to hypoxia and metabolic problems with a lower recovery capacity [25]. Complex shoulder injuries are predicted to have poor outcomes, with the hypoxic environment in smokers [21]. Based on our previous study, smoking impaired the vascularization of the biceps tendon in chronic tendinopathy, and we observed a negative correlation between smoking and neovascularization [9]. Surgical procedures are aimed at restoring the function of the torn tendons; however, the biology of the healing process needs to be preserved. We hypothesized that the negative impact of hazardous substances of smoke might be an important risk factor for poor regeneration and further impaired functional outcomes in complex shoulder injuries.

The purpose of this study was to investigate the association between smoking and functional outcomes after arthroscopic treatment of complex shoulder injuries, including rotator cuff tears with biceps tendon tears.

## 2. Materials and Methods

### 2.1. Bioethics

This study was performed following the Declaration of Helsinki for experiments involving humans, after receiving permission from local Bioethics Committee (approval number, KB 62/2017). All patients provided written informed consent before enrolling in the study.

### 2.2. Study Cohort

This retrospective case-control study was conducted on a cohort of patients who underwent shoulder arthroscopic surgery between 2015 and 2017 due to the treatment of the complex injury in the Department of Orthopedics. Inclusion criteria were: patients age ≥25 and <79 years, complex shoulder injury including LHBT and RC lesions, which were diagnosed preoperatively with physical examination and imaging techniques (sonography and magnetic resonance imaging) and confirmed intraoperatively (morphology of RCTs was determined as follows: variables A, B—partial RCTs, C—complete RCTs; variables 0–4 depending on range of tear, according to Snyder et al.), history of no improvement after minimum three months of the conservative treatment. A minimum 1-year follow–up was required for inclusion. Exclusion criteria were: SLAP tears, ACJ disorders, instability with an anterior or posterior labrum tear, prior humeral head fractures, and systemic inflammatory diseases. Demographic data including age, gender, comorbidities, operative factors, trauma, indications and smoking habits were recorded. The physical examination with dedicated tests for RC and LHBT disorders was done for each patient and the American Shoulder and Elbow Surgeons Score (ASES) was collected preoperatively and postoperatively.

### 2.3. Smoking Data

Smoking status was quantified by the following questions before treatment: have you ever smoked, how many years have you smoked, what is the average number of cigarettes per day, have you ever smoked after surgery. Furthermore, the pack-years index was calculated for each smoker.

### 2.4. Surgery

All repairs were performed arthroscopically by two authors (standard 30° arthroscope Smith&Nephew, systemic inflammatory diseases), in the beach-chair position with the repair technique and sutures depending on the configuration of the injury’s configuration. The standard posterior portal and additional working portals were used. Biceps tendon lesions were treated with tenotomy (the LHBT was first cut parallel to its insertion to the labral complex and the superior labrum was debrided) or tenodesis procedures (4.5 mm doubly loaded suture anchor, Twinfix, Smith&Nephew, Memphis, TN, USA), depending on surgeon experience. Rotator cuff tears were repaired using 4.5 mm suture anchors (Twinfix, Smith&Nephew, Memphis, TN, USA), depending on the morphology of lesions and the range of tear (Figure 1). The lesions diagnosed as type A or B (0–2) according to Snyder et al. were only debrided using arthroscopic shaver. Types A or B (3–4) and all of type C according to Snyder et al. were classified for surgical repair.

### 2.5. Rehabilitation

The rehabilitation protocol after shoulder arthroscopy in cases with partial-thickness RC repair (types A, B, 0–2, according to Snyder et al.) included an arm sling which had been worn for three weeks followed by subsequent exercises of range of motion and strength improvement. After shoulder arthroscopy with full-thickness RC repair (types A, B, 3–4, and type C according to Snyder et al. classification), arm abduction orthosis was used for five weeks, and rehabilitation was continued with range of motion (ROM) and strengthening exercises. Each patient obtained preoperative and postoperative information regarding a standard rehabilitation protocol.

### 2.6. Follow Up

The outcomes were assessed using the American Shoulder and Elbow Surgeons scale (ASES), the University of California at Los Angeles (UCLA) Shoulder Score, non-steroid anti-inflammatory drugs (NSAID) consumption, and the visual analog scale (VAS). Complications and changes in smoking status were also noted.

### 2.7. Statistical Analysis

The data were analyzed with the use of GraphPad Prism v.8.0.1 (GraphPad Software, La Jolla, CA, USA); *p* < 0.05 was considered significant. The data were compared with the non-parametric Mann-Whitney U test and Wilcoxon signed-rank test. Relations between the parameters studied were assessed using the Spearman correlation coefficient with use of correlation matrix function, and multiple linear regression analysis. The Chi-square test was used to compare descriptive characteristics. Two independent investigators performed all the comparisons between groups and statistical analyses.

## 3. Results

A total of 59 patients met the inclusion criteria and underwent shoulder arthroscopy due to complex LHBT pathology and RCTs. Demographic data and characteristics of patients were summarized in Table 1.

The mean age of patients at the time of surgery was 55.59 years; specifically, the mean age of smokers was 56 years, and non-smokers was 55.18 years. Gender distribution was 20 women to 39 men. According to smoking status, 27 patients were classified as smokers, and the remaining 32 were nonsmokers; however, seven patients in the smokers’ group were initially former smokers, who had quit smoking at least one month before surgery, but they admitted during follow-up that they did not completely quit the habit and occasionally smoked after surgery. This was the reason why they were included in the smokers’ group. None from smokers’ group gave up the habit in the follow-up period. Furthermore, the smokers’ group was predominated by men, and there were no women. The mean number of cigarettes smoked per day was 16.55 (range, 5–60; SD = 12.27), the mean period of smoking was 17.55 years (range, 5–30 years; SD = 8.46) and the mean pack-years index was 16.05 (range, 2.5–90; SD = 17.69) in this population.

In the examined cohort, 36 patients underwent the LHBT tenotomy and 23 tenodesis. Patients undergoing arthroscopic tenodesis (mean age 51.43, range: 27–75, SD—12.56) were statistically younger than those undergoing tenotomy (mean age 58.25, range: 38–73, SD—8.13), (*p* = 0.0093) (Figure 2A). Of the 59 shoulders that underwent arthroscopy, 11 (18.6%) were RCTs types A and B (0–2), 20 (33.8%) were RCTs types A and B (3–4), type C 1–3, and 28 (47%) were RCTs type C4; a total of 48 patients underwent cuff repair using suture anchors. The distribution of cuff tears morphology, depending on smoking status, was summarized in Figure 2B. We observed a relationship between smoking status and the distribution of various RCTs (*p* < 0.0001) (Figure 2B).

The mean follow-up in the examined population was 26.03 months; specifically, in smokers, it was mean 27.11 months, and non-smokers the mean was 25.12 months. There were no wound infections or reoperations reported during the follow-up period. The mean ASES score improved from 45.69 preoperatively to 82.57 postoperatively in the studied cohort (*p* < 0.0001) (Figure 2C). The mean ASES score improved from 45.11 to 80.81 in the smokers’ group and from 46.18 to 84.06 in the non-smokers’ group during the follow-up (*p* < 0.0001) (Figure 2D,E). The postoperative UCLA score distribution was 30.59 for the entire population, 30.18 and 30.93 for smokers and non-smokers’ groups, respectively. There were no statistically significant differences in pre/post ASES and postoperative UCLA score between smokers’ and non-smokers’ groups (*p* = 0.6785, *p* = 0.3143, *p* = 0.4946, respectively) (Figure 2F–H). There was no correlation between the age and pre/postoperative ASES, postoperative UCLA and follow up period (*p* = 0.6463, *p* = 0.6463, *p* = 0.0591, *p* = 0.1649, respectively).

Moreover, we isolated the group with the range of age from 33 to 66 and overall postoperative ASES was 81.9 (50–100; SD = 16.30), in non-smokers’ group: 83.57 (55–100; SD = 15.75) and in smokers’ group: 79.96 (50–100; SD = 17.06). The postoperative UCLA was 30.54 in studied population (20–35; SD = 4.49), specifically, in non-smokers’ group: 30.38 (21–35; SD = 4.67) and in smokers’ group: 30.46 (20–35; SD = 4.53).

The Popeye deformity and night pains were observed more frequently in the smokers’ group (*p* = 0.0002 and *p* = 0.1960, respectively) (Figure 3A,B). However, the rate of these complications could be biased by the tenotomy/tenodesis procedures, and it was explained in the limitations section. The mean postoperative VAS score was 1.66 and it was lower in the non-smokers’ group (0.96) vs. the smokers’ population (2.48) (*p* = 0.0021) (Figure 3C). The NSAID consumption was decreased in smokers’ population despite the increased mean VAS (*p* = 0.4127).

There was no correlation between pre/postoperative ASES, postoperative UCLA, and smoking indexes. However, there was a positive correlation between VAS and pack-year index (*p* < 0.0001), the number of cigarettes smoked per day (*p* = 0.0004), and smoking years (*p* < 0.0001) (Figure 3D–F). The correlation matrix using Spearman tests of all included variables was presented in Figure 4A,B, comparing the Spearman rho and *p* values, respectively.

Multiple linear regression analysis confirmed that smoking indexes, such as the number of cigarettes per day and pack-year index are useful in predicting the severity of rotator cuff injuries (*p* = 0.0155 and *p* = 0.0362 respectively) (Table 2; Table 3).

Moreover, the smoking years index is also statistically significant to predict the VAS outcome (*p* = 0.036) (Table 4).

## 4. Discussion

In the present study, we investigated whether tobacco smoking has an impact on mid-term functional results after arthroscopic treatment of complex shoulder injuries, which are rotator cuff with biceps tendon tears. It is a well-established fact that multi-tendon injuries of the shoulder are a serious challenge for surgeons, physiotherapists and it is difficult to reach a satisfying final functional outcome. Therefore, there is a need for a global consensus on shoulder multiligamentous injuries, presented by Oliva et al. in their multicenter study [26]. Chronic and irreparable massive rotator cuff tears remain one of the most challenging shoulder surgeries, and there is a wide spectrum of therapeutic options, with different degrees of invasiveness (muscle flaps, fascia lata transfer, partial repairs, tenotomy of the LHBT, superior capsular reconstruction, using a xenograft) [27]. Moreover, lesions exposed to the influence of harmful substances of tobacco smoke may have additionally disturbed the balance of the microenvironment of the tendinous tissue, resulting in an impaired healing process [9].

Since this one, the basic research interest about the influence of smoking on shoulder surgery has been primarily based on isolated rotator cuff tears examination. This topic was explored in a few studies [7,22,28]. However, we believed that LHBT is commonly involved in rotator cuff tears. This is such a strong relationship that both may interfere and cause significant tears of each other [15,16,17,18]. As far as we know, this is the first study that investigated the influence of smoking on outcomes after complex biceps and rotator cuff tendons surgery.

Until now, a few studies have investigated the influence of tobacco smoke on rotator cuff morbidity, surgery and clinical results. Bishop et al. analyzed thirteen studies that deal with RCTs and functional outcomes measured in clinical scales [29]. The authors concluded that smokers had an impaired functional outcome as well as more advanced tendon lesions. We have also revealed impaired postsurgical results in smokers’ group with LHBT and RCTs, compared to non-smokers and the tendency to massive C4 tears of the rotator cuff, according to Snyder, in the smoker group. McRae et al. showed that smoking was associated with poor shoulder function and higher pain scores, but there was no correlation with ASES scores, similar to our findings [30]. Furthermore, the measured UCLA score in our cohort and the general results were worse in the smoker group, akin to inferences of the study about isolated RCTs in smokers presented by Mallon et al. [20]. Fehringer et al. showed no statistically significant association between smoking and the prevalence of the RC tears [31]. We found a relationship between smoking status and the distribution of various RCTs. Still, the inclusion criteria in their study group were, e.g., age (>65 years), and it differentiates our study from the others. We included a younger population, because at the ages of 60, the prevalence of RCTs in the general population is relatively high, about >50% [32]. On the other hand, Leffa et al. presented the impact of smoking on physical therapy program after the RCTs arthroscopic treatment [33]. When compared to the non-smoking population, smokers had significantly increased pain in the shoulder during the final examination. Authors included into the study are mainly chronic smokers, with mean history of 30 years of smoking.

Reassuming smoking is a significant negative risk factor for rotator cuff pathology. However, the complexity of shoulder injury often leads to multiple-tendon injuries, mostly associated with the LHBT disorders. Authors have emphasized the role of simultaneous treatment of biceps and rotator cuff tendons to achieve an excellent clinical outcome, especially in older patients [17,34]. The LHBT disorders may occur insidiously and the symptoms are often ignored and assigned to RC pathology. Thus, a holistic approach can lead to satisfactory postoperative results [34,35]. Watson et al. evaluated patients with complex shoulder injuries: RCTs and LHBT, one year after shoulder arthroscopy [35]. Authors revealed that both tenotomy and tenodesis of the biceps tendon, as parts of rotator cuff repair, have ameliorated clinical outcomes measured in multiple clinical scales.

In contrast, isolated cuff repairs without the LHBT treatment had led to impaired functional results. Extensive arthroscopic treatment of multi-tendon lesions may be the key for improved clinical outcomes. LHBT lesions are important pain generators in the shoulder; missed diagnosis may lead to poor surgical outcomes and impaired rehabilitation process. In our study, the smoking population had worse surgical results. Nevertheless, there were no statistically significant differences when compared to non-smokers. Thus, we believe that holistic treatment of shoulder disorders could lead to an improvement in both groups.

Alicioglu et al. investigated achilles and patellar tendon pathology using sonography. The authors did not recognize any difference between smokers and non-smokers according to tendon thickness and incidence of tendinous degeneration [36]. Nevertheless, microscopic studies have presented rather different results revealing that degeneration observed in tendinous tissue of smokers is common and advanced [37,38,39]. Smoking causes microvascular disease of numerous tissues, resulting in impaired vascular supply and collagen synthesis and it seems to deteriorate the healing process [28]. Moreover, smoking increases the number of circulating proinflammatory cytokines, which may prolong inflammation and increase pain sensitivity. [29]. We showed that increased pain level had been measured in a dedicated scale postoperatively in the smoker group, an outcome similar to those of McRae et al. and Mallon et al. [20,30]. The rotator cuff and biceps tendons have hypovascular regions, making those tissues more prone to microtears, hypoxia, and imbalance in the healing process. Lundgreen et al. showed in their histopathological study of rotator cuff tendon samples derived from ten smokers that smoking is associated with early-onset tendon degeneration [37]. These samples were characterized by an advanced degenerative alteration with reduced tenocyte density and increased apoptotic cell density. Other authors investigated LHBT in the smoker population and indicated a more intensified degenerative process of the tendinous tissue [40]. These microscopic findings support the theory of impaired healing of tendinous tissue in smokers and the tendency to massive tears. This is another reason to treat these pathologies simultaneously because RC and LHBT are under the harmful influence of tobacco smoke substances, as shown in microscopic studies.

Popeye’s sign is a common complication after LHBT arthroscopic tenotomy. In this study, the higher rate of its occurrence was noted in the smokers’ group. Smokers have greater risk of biceps tendon rupture in proximal and distal parts of their attachment [41]. However, in this study the obtained data could be influenced by two inconsistent LHBT surgical procedures—tenotomy/tenodesis, which may have an impact on final results [42]. The smokers also had more often night pain, which also could be biased by the male/female ratio, since there were no females in the smoking population [43]. This field needs further research because there is no simple link between pain sensation and smoking habits [44].

This study has several limitations. For instance, we did not include a similar type of RCTs. There was a spectrum of tears morphology however we categorized them according to arthroscopic Snyder Classification. It allowed us to find an interesting distribution of various types. Two authors performed surgical techniques of tenodesis or tenotomy procedures chosen according to the surgeon’s preference, and the Popeye sign occurrence. Data shown in numerous studies revealed similar outcomes after these two surgical procedures. Furthermore, the smoker group was a quite differentiated population with various smoking indexes and various tobacco product consumption. It is rather difficult to measure the intake of harmful substances, but many authors noted this limitation. There was no MRI control; however, the functional outcomes were good and there was no re-operation during the follow-up. To minimize examiner dependency and bias during the follow-up, patients were evaluated by two surgeons independently.

## 5. Conclusions

Multi-tendon injuries of the shoulder are a serious challenge for surgeons, and to obtain an excellent functional outcome, we need to limit the negative risk factors, including smoking. Furthermore, there is a significant association between smoking and the occurrence of massive rotator cuff tears, and the pain level measured by the VAS. Simultaneous surgical treatment of RC and LHBT lesions in the smoker population allowed us to obtain the functional outcomes approximated to non-smokers in the long-term follow-up. Of course, we cannot assert that smoking is the real cause of all complications, however, we may assume that this is very important, negative factor in shoulder arthroscopy.

## Figures and Tables

**Figure 1 jcm-10-00599-f001:**
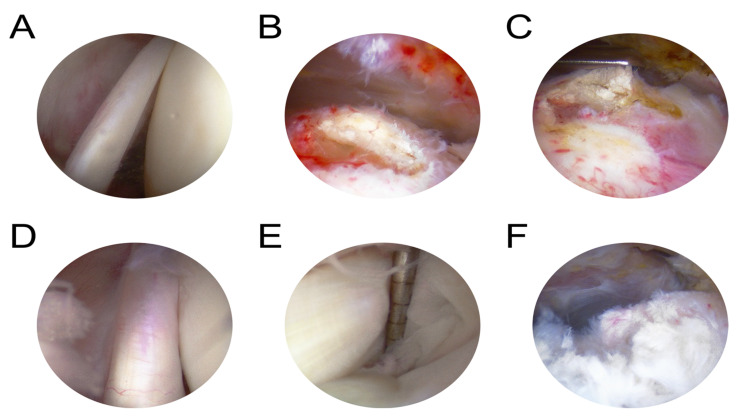
Arthroscopic slides obtained during the shoulder arthroscopy presenting biceps tendon pathology with abundant vascularization in (**A**,**D**); partial rotator cuff tears, articular side in (**E**) and bursal side in (**F**); and also, complete rotator cuff tears in (**B**,**C**).

**Figure 2 jcm-10-00599-f002:**
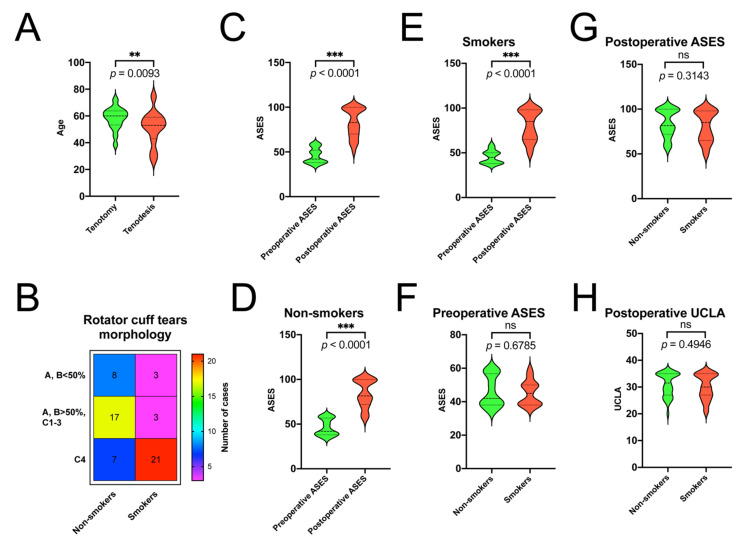
Summarized statistical analysis depending on age, the smoking status, rotator cuff tears morphology, pre/postoperative ASES and postoperative UCLA. (**A**) Comparison of age in tenotomy/tenodesis groups. (**B**) Heat map of distribution of RCTs morphology according to smoking status. (**C**) Comparison of preoperative ASES and postoperative ASES in general population. (**D**) Comparison of preoperative ASES and postoperative ASES in non-smoking population. (**E**) Comparison of preoperative ASES and postoperative ASES in smoking population. (**F**) Comparison of preoperative ASES in non-smoking and smoking population. (**G**) Comparison of postoperative ASES in non-smoking and smoking population. (**H**) Comparison of postoperative UCLA in non-smoking and smoking population; ns, *p* > 0.05; ** *p* > 0.0001; *** *p* < 0.0001.

**Figure 3 jcm-10-00599-f003:**
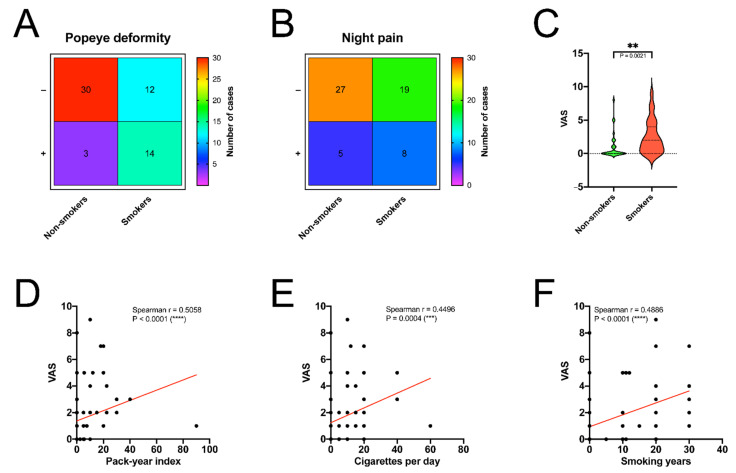
(**A**,**B**) Heat map of distribution of Popeye deformity and night pain according to smoking status; (**C**) Comparison of VAS in non-smoking and smoking population; (**D**,**E**,**F**) Correlation between the VAS and pack-years index, cigarettes per day, smoking years.

**Figure 4 jcm-10-00599-f004:**
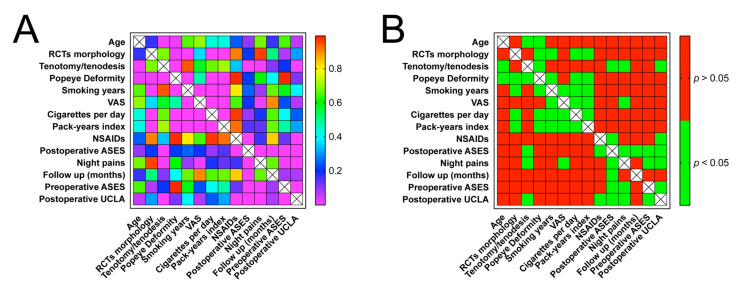
(**A**) Heat map of correlation matrix, according to Spearman rho; (**B**) Heat map of correlation matrix, according to *p*-value.

**Table 1 jcm-10-00599-t001:** Summary of demographic and clinical characteristics

Characteristics	Total	Smokers	Non-smokers
Mean Age	55.59 (27–75; SD = 10.53)	56 (27–74; SD = 10.07)	55.18 (28–75; SD = 11.05)
No. patients	59	27	32
Gender Distribution	20 women	0 women	20 women
	39 men	27 men	12 men
Mean follow up	26.03 (12–51; SD = 11.61)	27.11 (12–45; SD = 10.62)	25.12 (12–51; SD = 12.48)
Mean preoperative ASES	45.69 (34–60; SD = 8.43)	45.11 (35–60; SD = 7.37)	46.18 (34–60; SD = 9.32)
Mean postoperative ASES	82.57 (50–100; SD = 16.13)	80.81 (50–100; SD = 17.12)	84.06 (55–100; SD = 15.35)
Mean postoperative UCLA	30.59 (20–35; SD = 4.51)	30.18 (21–35; SD = 4.7)	30.93 (20–35; SD = 4.39)
Popeye deformity (+/−)	42 −	12 −	30 −
	17 +	14 +	3 +
Night pain (+/−)	13 +	8+	5+
NSAID (+/−)	14+	6+	8+
Mean VAS	1.66 (0–9, SD 2.29)	2.48 (0–9, SD 2.5)	0.96 (0–8; SD 1.87)

ASES—the American Shoulder and Elbow Surgeons Score, UCLA—the University of California at Los Angeles Shoulder Score, NSAIDs—non-steroid anti-inflammatory drugs, VAS—the visual analog scale.

**Table 2 jcm-10-00599-t002:** Results of multiple linear regression analysis of the relationships between cigarettes per day and other variables.

Cigarettes Per Day				
Variable	Estimate	SD	95% CI	*p* Value
Intercept	−14.00	8.915	−31.96 to 3.955	0.1233
Age	0.02295	0.06230	−0.1025 to 0.1484	0.7144
Complex	2.151	0.8548	0.4297 to 3.873	0.0155
Tenotomy/tenodesis	3.474	1.426	0.6031 to 6.345	0.0188
Popeye Deformity	4.347	1.581	1.163 to 7.530	0.0085
VAS	−0.01242	0.09006	−0.1938 to 0.1690	0.8909
Cigarettes per day	0.4204	0.2819	−0.1474 to 0.9882	0.1428
Pack-years index	0.6522	0.06481	0.5216 to 0.7827	<0.0001
NSAIDs	−0.8120	1.683	−4.202 to 2.578	0.6318
ASES postop.	−0.09665	0.08873	−0.2754 to 0.08206	0.2818
Night pains	−0.8049	1.900	-4.632 to 3.022	0.6739
Follow up (months)	0.06177	0.05544	−0.04989 to 0.1734	0.2711
ASES preop.	0.1302	0.08760	−0.04627 to 0.3066	0.1443
UCLA postop.	0.1537	0.3367	−0.5244 to 0.8319	0.6502

**Table 3 jcm-10-00599-t003:** Results of multiple linear regression analysis of the relationships between pack-years index and other variables.

Pack-Years Index				
Variable	Estimate	SD	95% CI	*p* Value
Intercept	14.76	11.47	−8.350 to 37.86	0.2049
Age	0.02987	0.07948	−0.1302 to 0.1900	0.7088
Complex	−2.394	1.109	−4.627 to -0.1614	0.0362
Tenotomy/tenodesis	−2.994	1.883	−6.786 to 0.7985	0.1188
Popeye Deformity	−2.773	2.140	−7.083 to 1.536	0.2016
VAS	0.3044	0.1056	0.09170 to 0.5170	0.0060
Cigarettes per day	−0.5246	0.3601	−1.250 to 0.2006	0.1521
Pack-years index	1.062	0.1055	0.8490 to 1.274	<0.0001
NSAIDs	0.7551	2.150	−3.575 to 5.085	0.7270
ASES postop.	0.04515	0.1145	−0.1854 to 0.2757	0.6952
Night pains	−0.1361	2.429	−5.028 to 4.756	0.9556
Follow up (months)	−0.08867	0.07047	−0.2306 to 0.05325	0.2147
ASES preop.	−0.1325	0.1127	−0.3596 to 0.09461	0.2462
UCLA postop.	−0.1230	0.4302	−0.9894 to 0.7434	0.7763

**Table 4 jcm-10-00599-t004:** Results of multiple linear regression analysis of the relationships between smoking years and other variables.

Smoking Years				
Variable	Estimate	SD	95% CI	*p* Value
Intercept	0.5434	15.15	−29.97 to 31.06	0.9716
Age	−0.1342	0.1013	−0.3382 to 0.06989	0.1921
Complex	1.169	1.501	−1.854 to 4.192	0.4401
Tenotomy/tenodesis	−0.1596	2.510	−5.215 to 4.896	0.9496
Popeye Deformity	1.219	2.821	−4.463 to 6.901	0.6678
VAS	0.9836	0.4549	0.06741 to 1.900	0.0360
Cigarettes per day	−0.03401	0.2467	−0.5308 to 0.4628	0.8909
Pack-years index	0.5122	0.1777	0.1543 to 0.8700	0.0060
NSAIDs	−1.366	2.785	−6.976 to 4.244	0.6263
ASES postop.	−0.02201	0.1487	−0.3216 to 0.2775	0.8830
Night pains	3.757	3.101	−2.488 to 10.00	0.2319
Follow up (months)	0.03271	0.09288	−0.1544 to 0.2198	0.7263
ASES preop.	0.08141	0.1480	−0.2167 to 0.3795	0.5850
UCLA postop.	0.1225	0.5582	−1.002 to 1.247	0.8273

## Data Availability

Data available on request due to restrictions eg privacy or ethical.

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
