# Peer review of "The Impact of Smoking on Clinical Results Following the Rotator Cuff and Biceps Tendon Complex Arthroscopic Surgery"

_jcm, 2021, doi:10.3390/jcm10040599_

Round 1
Reviewer 1 Report
1)The paper is a nice attemtp to demonstrate a different outcome in smokers with RCT and LHBT tear. Unluckely the limitations and the important bias cannot permit to reach a final conclusions.
2) I suggest to reduce the conclusions to what really the authors can realistic demonstrate. The association and the features of the smoking group is ok and well investigated but you cannot assert with different kind of lesions and surgical procedures that smoking is the real cause.
3) Please update the litterature regarding this issue.
4) Please P value when not significative type n.s.
5) Please add and discuss:
Cigarette smoking on the functional recovery of patients with rotator cuff tear submitted to physical therapy
Cardoso Leffa T., Galvão Novelli J., Biff dos Santos G., Maciel Bello G., da Silva Santos M., Martins Silveira M., Leindecker R.C., Boff Daitx R., Baptista Dohnert M. Muscles, Ligaments and Tendons Journal 2018 8(2) 150 - 162 doi: 10.32098/mltj.02.2018.03I.S.Mu.L.T – Rotator Cuff Tears Guidelines
Oliva F., Piccirilli E., Bossa M., Via A.G., Colombo A., Chillemi C., Gasparre G., Pellicciari L., Franceschetti E., Rugiero C., Scialdoni A., Vittadini F., Brancaccio P., Creta D., Del Buono A., Garofalo R., Franceschi F., Frizziero A., Mahmoud A., Merolla G., Nicoletti S., Spoliti M., Osti L., Padulo J., Portinaro N., Tajana G., Castagna A., Foti C., Masiero S., Porcellini G., Tarantino U., Maffulli N. Muscles, Ligaments and Tendons Journal 2015;5 (4):227-263 227 - 263 doi: 10.11138/mltj/2015.5.4.227Superior Capsule Reconstruction for Irreparable Rotator Cuff Tear with a Porcine Dermal Graft: Preliminary Results at 2 Years Minimum Follow-up
R. Dukan, A. Bommier, M. A. Rousseau, P. Boyer Muscles, Ligaments and Tendons Journal 2019;9 (4) doi: 10.32098/mltj.04.2019.12Author Response
Dear Editor,
Thank you for the opportunity to improve and resubmit our manuscript entitled:
“The impact of smoking on clinical results following the rotator cuff and biceps tendon complex arthroscopic surgery”
The suggestions offered by the reviewers have been immensely helpful. We appreciate all the comments on the manuscript.
We have included the reviewer comments, and responded to them individually, indicating how we addressed each concern and describing the changes we have made. The revised manuscript has been read and approved by all the authors.
We wish to express again our appreciation for the insightful comments which have helped us significantly to improve our manuscript.
Yours sincerely,
Jan Zabrzyński
Reviewer 1:
Overall, thank You for all the comments on the manuscript and good words.
Keyword: change “massive and irreparable rotator cuff tears” in “massive rotator cuff tears.”
Thank you for the comment. This mistake was corrected.
Introduction:
Line 44 – Authors stated that smoking impacts the musculoskeletal system after various surgical procedures. Which procedures? Please add a proper reference. References 1-3 is related only to shoulder surgery and not to “various procedures”.
Thank you for the comment. This mistake was corrected, and specific procedures are mentioned, such as: knee arthroplasty, meniscus surgery and others. The references were also modified according to a new data.
Methods
The age range is too wide (25-79 years). Risk factors for rotator cuff injury and recurrence could vary during the ageing. Young people could be injured by traumatic mechanism, instead of in older patients, the degenerative mechanism could affect the shoulder. I suggest to perform the analysis layering the population by age (maybe 25-50; 50-75 or similar).
Thank you for the comment, we fully agree with You.
We isolated the group with range of age from 33 to 66 and overall postoperative ASES was 81.9 (50-100; SD=16.30), in non-smokers group: ASES 83.57 (55-100; SD=15.75) and in smokers group: 79.96 ( 50-100; SD=17.06). Moreover, the postoperative UCLA was 30.54 (20-35; SD=4.49), in non-smokers group: 30.38 (21-35; SD=4.67) and in smokers group: 30.46 (20-35; SD=4.53).
These results are almost identical as in the primary: 25-79 years population. This data was also included into main text.
Did the same equipe perform the surgery? Different outcomes could be related to different surgeons.
Thank you for the valuable comment – the same team, two authors (JZ, ŁP) performed the shoulder arthroscopy.
Authors performed tenodesis and tenotomies. This could constitute an important bias regarding the insurgence of popeye sign! Tenotomy is associated with a higher risk of Popeye sign compared to tenodesis.
(Meta-Analysis J Orthop Surg Res . 2019 Nov 15;14(1):370. doi: 10.1186/s13018-019-1429-x.
A meta-analysis comparing tenotomy or tenodesis for lesions of the long head of the biceps tendon with concomitant reparable rotator cuff tears
Yuyan Na 1, Yong Zhu 2, Yuting Shi 3, Yizhong Ren 1, Ting Zhang 1, Wanlin Liu 4, Changxu Han 5
PMID: 31729995 PMCID: PMC6858715 DOI: 10.1186/s13018-019-1429-x
This bias could influence the results you reported. I suggest to delete the popeye signs’ analysis or insert this limitation in the discussion.
Thank you for the valuable comment and we decided to explain it well in the results and limitations sections to reduce the risk of bias associated with tenotomy/tenodesis procedures and the rate of Popeye deformation.
No women were enrolled in the smoker’s group. Sex could influence postoperative outcomes. For example, Women reported more significant pain and decreased shoulder function compared with men during the initial three months after arthroscopic rotator cuff repair. Please also add this limitation.
Orthop J Sports Med . 2019 Nov 25;7(11):2325967119881959. doi: 10.1177/2325967119881959. eCollection 2019 Nov.
Sex-Based Differences in Patient-Reported Outcomes After Arthroscopic Rotator Cuff Repair
Stephen D Daniels 1 2, Cory M Stewart 1, Kirsten D Garvey 1, Emily M Brook 1, Laurence D Higgins 1 3, Elizabeth G Matzkin 1 3
PMID: 31803785 PMCID: PMC6878615 DOI: 10.1177/2325967119881959
Thank you for the valuable comment and we decided to include this important limitation.
Results
All the results reported could be influenced by the previous problems described (tenodesis/tenotomy and lack of female subjects)
Line 205 – authors stated NSAID was decreased in smokers. Please specify that this difference is not significant (p=0.23)
Thank you for the valuable comment and the p-value was filled in the text.
Discussion
Line 239 – Please add ref.
Thank you for the valuable comment, the literature was updated.
Line 304 – Popeye’s sign is a common complication. Basing on previous comments, authors cannot state that.
Line 308 – Smokers had more often night pain. Basing on previous comments, authors cannot state that.
Thank you for the valuable comment, according to modification from methods section, these paragraphs were intensively modified due to important limitations.
Conclusions
Line 326 – Please modify the conclusions accordingly.
I suggest to add a separated limitations section or to perform a new analysis with a less heterogeneous group (including females in both groups, layering by age and including only tenotomy or tenodesis)
Best regards
Thank you for the valuable comment, the conclusions section was transformed according to Your advices.
Reviewer 2:
1)The paper is a nice attemtp to demonstrate a different outcome in smokers with RCT and LHBT tear. Unluckely the limitations and the important bias cannot permit to reach a final conclusions.
Thank you for the valuable comment, the conclusions section was transformed according to Reviewers advices.
2) I suggest to reduce the conclusions to what really the authors can realistic demonstrate. The association and the features of the smoking group is ok and well investigated but you cannot assert with different kind of lesions and surgical procedures that smoking is the real cause.
Thank you for the valuable comment, the conclusions section was transformed according to Reviewers advices.
3) Please update the litterature regarding this issue.
Thank you for the valuable comment, the literature was updated.
4) Please P value when not significative type n.s.
Thank you for the valuable comment, it was improved.
5) Please add and discuss
Thank you for the valuable comment, the paper were incorporated into the discussion section.
Reviewer 2 Report
Dear authors,
I read the work of Zabrzyński and colleagues. The topic is interesting, but it is not so actual. The work is well written, and the methodology is correct. Otherwise, I have some concerns regarding your results and conclusions.
I have made some major revisions.
Keyword: change “massive and irreparable rotator cuff tears” in “massive rotator cuff tears.”
Introduction:
Line 44 – Authors stated that smoking impacts the musculoskeletal system after various surgical procedures. Which procedures? Please add a proper reference. References 1-3 is related only to shoulder surgery and not to “various procedures”.
Methods
The age range is too wide (25-79 years). Risk factors for rotator cuff injury and recurrence could vary during the ageing. Young people could be injured by traumatic mechanism, instead of in older patients, the degenerative mechanism could affect the shoulder. I suggest to perform the analysis layering the population by age (maybe 25-50; 50-75 or similar).
Did the same equipe perform the surgery? Different outcomes could be related to different surgeons.
Authors performed tenodesis and tenotomies. This could constitute an important bias regarding the insurgence of popeye sign! Tenotomy is associated with a higher risk of Popeye sign compared to tenodesis.
(Meta-Analysis J Orthop Surg Res . 2019 Nov 15;14(1):370. doi: 10.1186/s13018-019-1429-x.
A meta-analysis comparing tenotomy or tenodesis for lesions of the long head of the biceps tendon with concomitant reparable rotator cuff tears
Yuyan Na 1, Yong Zhu 2, Yuting Shi 3, Yizhong Ren 1, Ting Zhang 1, Wanlin Liu 4, Changxu Han 5
PMID: 31729995 PMCID: PMC6858715 DOI: 10.1186/s13018-019-1429-x
This bias could influence the results you reported. I suggest to delete the popeye signs’ analysis or insert this limitation in the discussion.
No women were enrolled in the smoker’s group. Sex could influence postoperative outcomes. For example, Women reported more significant pain and decreased shoulder function compared with men during the initial three months after arthroscopic rotator cuff repair. Please also add this limitation.
Orthop J Sports Med . 2019 Nov 25;7(11):2325967119881959. doi: 10.1177/2325967119881959. eCollection 2019 Nov.
Sex-Based Differences in Patient-Reported Outcomes After Arthroscopic Rotator Cuff Repair
Stephen D Daniels 1 2, Cory M Stewart 1, Kirsten D Garvey 1, Emily M Brook 1, Laurence D Higgins 1 3, Elizabeth G Matzkin 1 3
PMID: 31803785 PMCID: PMC6878615 DOI: 10.1177/2325967119881959
Results
All the results reported could be influenced by the previous problems described (tenodesis/tenotomy and lack of female subjects)
Line 205 – authors stated NSAID was decreased in smokers. Please specify that this difference is not significant (p=0.23)
Discussion
Line 239 – Please add ref
Line 304 – Popeye’s sign is a common complication. Basing on previous comments, authors cannot state that.
Line 308 – Smokers had more often night pain. Basing on previous comments, authors cannot state that.
Conclusions
Line 326 – Please modify the conclusions accordingly.
I suggest to add a separated limitations section or to perform a new analysis with a less heterogeneous group (including females in both groups, layering by age and including only tenotomy or tenodesis)
Best regards
Author Response
Dear Editor,
Thank you for the opportunity to improve and resubmit our manuscript entitled:
“The impact of smoking on clinical results following the rotator cuff and biceps tendon complex arthroscopic surgery”
The suggestions offered by the reviewers have been immensely helpful. We appreciate all the comments on the manuscript.
We have included the reviewer comments, and responded to them individually, indicating how we addressed each concern and describing the changes we have made. The revised manuscript has been read and approved by all the authors.
We wish to express again our appreciation for the insightful comments which have helped us significantly to improve our manuscript.
Yours sincerely,
Jan Zabrzyński
Reviewer 1:
Overall, thank You for all the comments on the manuscript and good words.
Keyword: change “massive and irreparable rotator cuff tears” in “massive rotator cuff tears.”
Thank you for the comment. This mistake was corrected.
Introduction:
Line 44 – Authors stated that smoking impacts the musculoskeletal system after various surgical procedures. Which procedures? Please add a proper reference. References 1-3 is related only to shoulder surgery and not to “various procedures”.
Thank you for the comment. This mistake was corrected, and specific procedures are mentioned, such as: knee arthroplasty, meniscus surgery and others. The references were also modified according to a new data.
Methods
The age range is too wide (25-79 years). Risk factors for rotator cuff injury and recurrence could vary during the ageing. Young people could be injured by traumatic mechanism, instead of in older patients, the degenerative mechanism could affect the shoulder. I suggest to perform the analysis layering the population by age (maybe 25-50; 50-75 or similar).
Thank you for the comment, we fully agree with You.
We isolated the group with range of age from 33 to 66 and overall postoperative ASES was 81.9 (50-100; SD=16.30), in non-smokers group: ASES 83.57 (55-100; SD=15.75) and in smokers group: 79.96 ( 50-100; SD=17.06). Moreover, the postoperative UCLA was 30.54 (20-35; SD=4.49), in non-smokers group: 30.38 (21-35; SD=4.67) and in smokers group: 30.46 (20-35; SD=4.53).
These results are almost identical as in the primary: 25-79 years population. This data was also included into main text.
Did the same equipe perform the surgery? Different outcomes could be related to different surgeons.
Thank you for the valuable comment – the same team, two authors (JZ, ŁP) performed the shoulder arthroscopy.
Authors performed tenodesis and tenotomies. This could constitute an important bias regarding the insurgence of popeye sign! Tenotomy is associated with a higher risk of Popeye sign compared to tenodesis.
(Meta-Analysis J Orthop Surg Res . 2019 Nov 15;14(1):370. doi: 10.1186/s13018-019-1429-x.
A meta-analysis comparing tenotomy or tenodesis for lesions of the long head of the biceps tendon with concomitant reparable rotator cuff tears
Yuyan Na 1, Yong Zhu 2, Yuting Shi 3, Yizhong Ren 1, Ting Zhang 1, Wanlin Liu 4, Changxu Han 5
PMID: 31729995 PMCID: PMC6858715 DOI: 10.1186/s13018-019-1429-x
This bias could influence the results you reported. I suggest to delete the popeye signs’ analysis or insert this limitation in the discussion.
Thank you for the valuable comment and we decided to explain it well in the results and limitations sections to reduce the risk of bias associated with tenotomy/tenodesis procedures and the rate of Popeye deformation.
No women were enrolled in the smoker’s group. Sex could influence postoperative outcomes. For example, Women reported more significant pain and decreased shoulder function compared with men during the initial three months after arthroscopic rotator cuff repair. Please also add this limitation.
Orthop J Sports Med . 2019 Nov 25;7(11):2325967119881959. doi: 10.1177/2325967119881959. eCollection 2019 Nov.
Sex-Based Differences in Patient-Reported Outcomes After Arthroscopic Rotator Cuff Repair
Stephen D Daniels 1 2, Cory M Stewart 1, Kirsten D Garvey 1, Emily M Brook 1, Laurence D Higgins 1 3, Elizabeth G Matzkin 1 3
PMID: 31803785 PMCID: PMC6878615 DOI: 10.1177/2325967119881959
Thank you for the valuable comment and we decided to include this important limitation.
Results
All the results reported could be influenced by the previous problems described (tenodesis/tenotomy and lack of female subjects)
Line 205 – authors stated NSAID was decreased in smokers. Please specify that this difference is not significant (p=0.23)
Thank you for the valuable comment and the p-value was filled in the text.
Discussion
Line 239 – Please add ref.
Thank you for the valuable comment, the literature was updated.
Line 304 – Popeye’s sign is a common complication. Basing on previous comments, authors cannot state that.
Line 308 – Smokers had more often night pain. Basing on previous comments, authors cannot state that.
Thank you for the valuable comment, according to modification from methods section, these paragraphs were intensively modified due to important limitations.
Conclusions
Line 326 – Please modify the conclusions accordingly.
I suggest to add a separated limitations section or to perform a new analysis with a less heterogeneous group (including females in both groups, layering by age and including only tenotomy or tenodesis)
Best regards
Thank you for the valuable comment, the conclusions section was transformed according to Your advices.
Reviewer 2:
1)The paper is a nice attemtp to demonstrate a different outcome in smokers with RCT and LHBT tear. Unluckely the limitations and the important bias cannot permit to reach a final conclusions.
Thank you for the valuable comment, the conclusions section was transformed according to Reviewers advices.
2) I suggest to reduce the conclusions to what really the authors can realistic demonstrate. The association and the features of the smoking group is ok and well investigated but you cannot assert with different kind of lesions and surgical procedures that smoking is the real cause.
Thank you for the valuable comment, the conclusions section was transformed according to Reviewers advices.
3) Please update the litterature regarding this issue.
Thank you for the valuable comment, the literature was updated.
4) Please P value when not significative type n.s.
Thank you for the valuable comment, it was improved.
5) Please add and discuss
Thank you for the valuable comment, the paper were incorporated into the discussion section.
Round 2
Reviewer 2 Report
Authors modified the text according to the suggestions.